# FESNet: Frequency-Enhanced Saliency Detection Network for Grain Pest Segmentation

**DOI:** 10.3390/insects14020099

**Published:** 2023-01-17

**Authors:** Junwei Yu, Fupin Zhai, Nan Liu, Yi Shen, Quan Pan

**Affiliations:** 1School of Artificial Intelligence and Big Data, Henan University of Technology, Zhengzhou 450001, China; 2College of Information Science and Engineering, Henan University of Technology, Zhengzhou 450001, China; 3Basis Department, PLA Information Engineering University, Zhengzhou 450001, China; 4School of Automation, Northwestern Polytechnical University, Xi’an 710129, China

**Keywords:** visual saliency, grain pest segmentation, discrete wavelet transform, discrete cosine transform, deep frequency feature

## Abstract

**Simple Summary:**

Insect pests cause major nutritional and economic losses in stored grains through their pestilential activities, such as feeding, excretion, and reproduction. Therefore, the detection of grain pests and the estimation of their population density are necessary for taking the proper management initiatives in order to control insect infestation. The popular techniques for the detection of grain pests include probe sampling, acoustic detection, and image recognition, among which the image recognition can provide rapid, economic and accurate solutions for the detection of grain pests. With the development of deep learning, convolutional neural networks (CNN) have been extensively used in image classification and object detection. Nevertheless, the pixel-level segmentation of small pests from the cluttered grain background remains a challenging task in the detection and monitoring of grain pests. Inspired by the observation that humans and birds can find the insects in grains with a glance, we propose a saliency detection model to detect the insects in pixels. Firstly, we construct a dedicated dataset, named GrainPest, with small insect objects in realistic storage scenes. Secondly, frequency clues for both the discrete wavelet transformation (DWT) and the discrete cosine transformation (DCT) are leveraged to enhance the performance of salient object segmentation. Moreover, we design a new receptive field block, aggregating multiscale saliency features to improve the segmentation of small insects.

**Abstract:**

As insect infestation is the leading factor accounting for nutritive and economic losses in stored grains, it is important to detect the presence and number of insects for the sake of taking proper control measures. Inspired by the human visual attention mechanism, we propose a U-net-like frequency-enhanced saliency (FESNet) detection model, resulting in the pixelwise segmentation of grain pests. The frequency clues, as well as the spatial information, are leveraged to enhance the detection performance of small insects from the cluttered grain background. Firstly, we collect a dedicated dataset, GrainPest, with pixel-level annotation after analyzing the image attributes of the existing salient object detection datasets. Secondly, we design a FESNet with the discrete wavelet transformation (DWT) and the discrete cosine transformation (DCT), both involved in the traditional convolution layers. As current salient object detection models will reduce the spatial information with pooling operations in the sequence of encoding stages, a special branch of the discrete wavelet transformation (DWT) is connected to the higher stages to capture accurate spatial information for saliency detection. Then, we introduce the discrete cosine transform (DCT) into the backbone bottlenecks to enhance the channel attention with low-frequency information. Moreover, we also propose a new receptive field block (NRFB) to enlarge the receptive fields by aggregating three atrous convolution features. Finally, in the phase of decoding, we use the high-frequency information and aggregated features together to restore the saliency map. Extensive experiments and ablation studies on our dataset, GrainPest, and open dataset, Salient Objects in Clutter (SOC), demonstrate that the proposed model performs favorably against the state-of-the-art model.

## 1. Introduction

Grains, including cereals, oilseeds, and legumes, provide humans and livestock with a majority of food and other nutritional materials. Insect infestation is one of the leading factors accounting for the postharvest loss of grains during storage, and it is paramount to the nutritional status and economy of many countries [1]. According to statistics, the stored-grain losses, due to insect infestation, account for 6–10% of the total grains output annually in the world, while these damaged grains are equivalent to the annual ration of 200 million people [2].

At present, stored-grain pests are mainly detected by probe sampling, acoustic detection, and image recognition [3]. The probe sampling method is labor-intensive, time-consuming and difficult to satisfy the demand of grain storage modernization. The acoustic techniques can be used to estimate the type and density of insects by analyzing the sound of insects’ movement and feeding [3]. This method needs to separate the specific frequency sound of pests from the background sound signal, which is greatly affected by ambient noises in real storage scenes. The high cost and sensor sensitivity also limit the applicability of acoustic devices.

The image recognition-based methods comprehensively use image processing, feature description, machine learning, and other related technologies to detect and identify stored-grain pests. These methods have the advantages of using simple equipment, high accuracy, good consistency and flexible application, and have attracted many researchers’ attention in recent years. A lot of image-based methods are proposed for the detection and recognition of grain pests. For example, Shi et al. [4] proposed an improved detection network architecture based on R-FCN, in order to facilitate the detection and classification of eight common stored-grain pests. Lyu et al. [5] proposed a feature fusion SSD algorithm based on the top-to-down strategy for grain pest detection. In order to implement intelligent monitoring for insects in grain warehouses, Li et al. [6] constructed a feature pyramid network to extract multi-scale features for insect classification and bounding box regression. Chen et al. [7] designed a monitoring car that can run on the granary surface and an embedded model, based on YOLOv4, that can detect and identify two typical stored-wheat pests: the red flour beetle and rice weevil.

Humans and birds have a perfect visual ability to search for insects in many grains. Salient object detection (SOD) can simulate this biological vision mechanism to locate and extract the pixel-level segmentation of the most attractive objects in a scene [8]. A great deal of research works, based on convolutional neural network (CNN), have been focused on saliency detection in the last several decades. Most of the SOD models and datasets assume that each image contains one or more large and clear salient object. Grain pest segmentation in realistic storage scenes brings great challenges, because the objects of insects are small and the backgrounds of different grains are cluttered. The realistic image of more salient targets or non-salient objects in non-infested grains make insect saliency detection more challenging. Existing deep-learning-based models of SOD regularly perform convolution in the spatial domain. The pooling operations (max-pooling, average-pooling) in convolution down-sampling amplify random noise and reduce the semantic information of the image; this affects the segmentation accuracy of salient objects. As the existing methods cannot achieve satisfactory results for stored-grain pest detection, can other clues, such as frequency information, be considered to improve saliency detection accuracy? This paper focuses on the convolution in the frequency domain and proposes a frequency-enhanced saliency detection model (FESNet) special for stored-grain pest segmentation.

The main contributions of our work are summarized as follows.

(1) We construct a new and high-quality saliency detection dataset, named GrainPest. GrainPest contains 500 images with the attributes of small object, the appearance change of different insects, and the cluttered background of different grains. GrainPest provides a dataset benchmark for the subsequent stored-grain pest detection, as well as agricultural and forestry pest control.

(2) We propose a frequency-enhanced saliency detection network (FESNet) with the discrete wavelet transformation (DWT) and the discrete cosine transformation (DCT), involved in the traditional convolution layers. This model uses the DWT to improve the high-frequency information of the network from the perspective of the frequency domain, and uses the DCT to enhance channel attention and thus enrich low-frequency information. Then, we design a new receptive field block (NRFB) to aggregate multi-scale features to enlarge the global receptive field.

(3) Extensive experiments and ablation studies are carried out on the proposed dataset, GrainPest, and the open dataset, SOC. We compare FESNet with four State-of-the-Art (SOTA) SOD models. Experiment results for the metrics of S-measure, E-measure, Mean Absolute Error (MAE) and maximum F-measure show that FESNet performs extraordinarily well in the detection of small objects in cluttered backgrounds.

## 2. Related Works

### 2.1. Salient Object Detection in the Spatial Domain

Saliency object detection (SOD) refers to the simulation of human visual characteristics through certain algorithms, allowing the computer to extract the most attractive object in the image and then segment its pixel-level contour [9]. The spatial domain, also called image space, refers to the space composed of image pixels. In image space, spatial domain processing refers to the direct processing of the pixel values, with length as the independent variable. In recent years, many image-based methods for saliency object detection have been proposed [10,11,12]. The early SOD algorithm [13] works mainly based on the handcrafted saliency map for feature prediction. Recently, the convolutional neural network (CNN) has made new progress in saliency detection, due to its power for feature representation. As one of the most representative networks, Qin, X. et al. [14] proposed a two-level nested U-structure network (U^2^-Net) that can effectively combine the low and high-level features through the ReSidual U-blocks (RSU). U^2^-Net increased the depth of the network without increasing the computational cost, considering the pooling operations in the RSU, and achieved excellent results for salient object segmentation.

### 2.2. Salient Object Detection in the Frequency Domain

The frequency domain describes image features with spatial frequency as an independent variable. As a powerful tool for signal processing, frequency analysis has been widely used in the field of deep learning in recent years. Gueguen et al. [15] classified images by extracting features from the frequency domain. Xu et al. [16] found that CNN models are more sensitive to low and high-frequency information by analyzing image classification, detection, and segmentation tasks in the frequency domain. Qin, Z. et al. [17] proved that global average pooling was a special case for the decomposition of features in the frequency-domain. They proposed a multi-spectral channel attention mechanism, named FcaNet, which achieved state-of-the-art results in many tasks, such as image classification, object detection, and instance segmentation.

Different from the Fourier transform, whose basic function is a sine function, the wavelet works based on some small waves, featuring variable frequency and a limited duration [18]. The series expansion with the DWT will result in a discrete function, which refers to the discretization of scale and displacement parameters. The DWT has been widely applied in image research. Gunaseelan et al. [19] proposed that the DWT could be used to decompose the equalized image into frequency sub-bands and give interpolation to improve the contrast of the image resolution. Jee et al. [20] proposed that the DWT could be used for digital images, with the threshold processing algorithm adopted to remove noise in sub-bands.

The discrete cosine transform (DCT) is a special form of the discrete Fourier transform, with the cosine function as the transform kernel [21]. The DCT is not only characterized by orthogonal transformation, but also by the basis vector of its transformation matrix that can well describe the correlation characteristics of human voice and image signals. The DCT is mainly used for data or image compression. It can transform spatial domain signals into the frequency domain, and performs well in decorrelation. Ulicny et al. [22] proposed the DCT-based harmonic block to replace the traditional convolutional layer and used the DCT characteristics of energy compression to effectively compress the high-frequency information of the network. Abadi et al. [23] used the DCT to preprocess the information and insert it into the optimal range, after the image color component was converted into a wavelet domain.

Recently, as can be seen from the research, saliency object detection, based on the frequency domain, has shown more competitive performance; for example, Liu et al. [24] proposed a multi-level wavelet CNN model to reduce the size of the feature map in the shrinking subnet through wavelet transform and deploy an inverse wavelet transform to reconstruct the high-resolution feature map. Zhong et al. [25] performed the DWT and IDWT on features in the CNN to reconstruct high-resolution images with better textural details. Li et al. [26] integrated the frequency and spatial features into each convolutional layer, in order to capture salient objects complete in structure and clear in the boundary.

### 2.3. Salient Object Detection Dataset

With large-scale datasets and deep network structures proposed and continuously updated, saliency object detection has made great progress, but small object detection in a cluttered background is still a challenging problem. The popular saliency detection datasets, such as DUTS-TR [27], DUTS-TE [27], THUR-B [28], PASCAL-S [29], HKU-IS [30], SOD [31], MSRA [32] and ECSSD [33], assume that each image contains only one or two large salient objects, and that the objects are mostly located in the center of each image. These datasets ignore the small objects and non-salient objects in real-world scenes. Fan et al. [34] collected a dataset named Salient in Clutter (SOC), which contains 6000 images; half of them are pure texture images with non-salient objects. We considered small insects and non-salient images with pure grain backgrounds when constructing the dedicated dataset, GrainPest. As shown in Figure 1, we calculated the ratio between the salient objects and the image area as the scale attribute of common saliency detection datasets with formula (1). The image was defined as containing small salient objects if the area ratio was no larger than 10%. It can be seen that the proportion of small objects in both the GrainPest dataset and the SOC dataset [34] is more than 65%.

GrainPest, a dataset specially constructed for stored-grain pests, includes 57.4% small object images and 16% images without any salient target, so it is a typical small object dataset. However, other datasets rarely consider the situation without salient objects, but mostly consider mostly medium or large-sized objects.
(1)$$R=\frac{\sum_{x=1}^{W}\sum_{y=1}^{H}\left(pix(x,y)=1\right)}{W\times H}$$
where R is the area ratio of the salient object size in the image, W and H indicate the width and height of the image, respectively, and pix(x,y)=1 shows that the pixel belongs to a salient object. According to the ratio R of salient objects, we divide salient objects into four grades: H1(*R* ≤ 10%), H2(10% < R ≤ 20%), H3(20% < R ≤ 30%) and H4(R > 30%).

## 3. Materials and Methods

In this section, we first analyze the theory of the DWT and the DCT in the frequency domain. Then, based on the work above, we elaborate on the network structure of the saliency detection model for stored-grain pests, based on the frequency domain.

### 3.1. Discrete Wavelet Transform

The DWT focuses on using different filters (mainly high-pass and low-pass) to analyze different frequency signals. The analysis in this section mainly targets 1D/2D Haar wavelets, but it can be applied to other wavelets if only a small change is made.

The 1D Haar wavelets are the simplest wavelet basis function. The DWT uses the low-pass and high-pass filters of the Haar wavelet transform to decompose the original 1D data into low-frequency and high-frequency components. The low-frequency information corresponds to the average value and stores the image contour information; the high-frequency information corresponds to the difference value and stores the image details. On the contrary, the IDWT uses the same filter to restore the original data’s decomposed low-frequency and high-frequency components.

Where Haar scaling function Φ:(2)$$\Phi \left(x\right)=\left\{ \begin{array}{l} 1,\;for\;0\leq x<1\\ 0,\;otherwise \end{array}\right.$$
(3)$$\Phi \begin{array}{c} j\\ i \end{array}\left(x\right)=\Phi (2^{j}x-i)i=0,\ldots ,2^{j}-1.$$

Haar wavelet function φ:(4)$$\varphi (x)=\left\{ \begin{array}{l} 1,\;\;\;\;for\;0\leq x<1/2\\ -1,\;for\;1/2\leq x<1\\ 0,\;\;\;\;otherwise \end{array}\right.$$
(5)$$\varphi_{x}^{j}(x)=\varphi (2^{j}x-i)\;i=0,\ldots ,2^{j}$$

$2^{j}$ indicates the compression ratio, while i is the unit for displacement.

Furthermore, 2D data is intuitively reflected as images. The 2D DWT is essentially a discrete matrix for processing 2D numerical values. The 2D DWT performs low-pass and high-pass filtering in both horizontal and vertical directions to decompose 2D data into one low-frequency component and three high-frequency components. The low-frequency component is a low-resolution version containing the main information of the data. High-frequency components store vertical, horizontal, and diagonal details of data, respectively. Similarly, the 2D IDWT can reconstruct the original data information based on such components.

In general, high-dimensional wavelet filters are tensor products of 1D wavelet filters; for example, the low-pass and high-pass filters of the 1D Haar wavelet are
(6)$$f_{\;l}^{H}=\frac{1}{\sqrt{2}}{\left(1,1\right)}^{T},{\;f}_{\;h}^{H}=\frac{1}{\sqrt{2}}{\left(1,-1\right)}^{T}$$
where $f_{l}$ indicates a low-pass filter and $f_{h}$ indicates a high-pass filter.

The corresponding 2D Haar wavelet filter is
(7)$$f_{\;ll}^{H}=f_{\;l}^{H}⊗f_{\;l}^{H}=\frac{1}{2}[\begin{array}{l} 1\\ 1 \end{array}\begin{array}{l} 1\\ 1 \end{array}]$$
(8)$$f_{\;hl}^{H}=f_{\;l}^{H}⊗f_{\;h}^{H}=\frac{1}{2}[\begin{array}{l} 1\\ 1 \end{array}\begin{array}{l} -1\\ -1 \end{array}]$$
(9)$$f_{\;lh}^{H}=f_{\;h}^{H}⊗f_{\;l}^{H}=\frac{1}{2}[\begin{array}{l} 1\\ -1 \end{array}\begin{array}{l} 1\\ -1 \end{array}]$$
(10)$$f_{\;hh}^{H}=f_{\;h}^{H}⊗f_{\;h}^{H}=\frac{1}{2}[\begin{array}{l} 1\\ -1 \end{array}\begin{array}{l} -1\\ 1 \end{array}]$$
where $f_{ll}$ is a low-pass filter, while $f_{lh}$,$f_{hl}$ and $f_{hh}$ are high-pass filters of 2D Haar wavelets.

### 3.2. Discrete Cosine Transform

The DCT is essentially a discrete Fourier transform (DFT) whose input signal is a real even function. It can transform signals in the spatial domain into the frequency domain and is mainly used to compress data or images. During image compression and quantification, the DCT coefficient has its energy mainly concentrated in the upper left corner after the original image DCT. The transformed DCT coefficients that are less than a certain value are reset to zero by threshold operation. Then, a compressed image can be obtained via IDCT operation.

The 1D DCT transform has eight patterns; here we will only discuss the most commonly used second pattern. It can be expressed as follows:(11)$$F(u)=c(u)\sum_{i=0}^{N-1}f(i)\cos\left[\frac{(i+0.5)\pi }{N}u\right]$$
(12)$$c(u)=\left\{ \begin{array}{l} \sqrt{\frac{1}{N},}\;u=0\\ \sqrt{\frac{2}{N},}\;u\neq 0 \end{array}\right.$$
where f(i) is the original signal, F(u) is the coefficient after the DCT transform, N is the number of the original signals, and c(u) can be considered as a compensation coefficient, which can make the DCT transform matrix become an orthogonal matrix.

The 2D DCT transform means that the DCT transform is doubled, based on the 1D DCT transform. The formula is as follows:(13)$$F(u,v)=c(u)c(v)\sum_{i=0}^{N-1}\sum_{j=0}^{N-1}f(i,j)\cos\left[\frac{(i+0.5)\pi }{N}u\right]\cos\left[\frac{(j+0.5)\pi }{N}v\right]$$

### 3.3. Architecture of FESNet

Our FESNet adopts Encoder–Decoder architecture and selects ResNet [35] as the backbone network. However, in the network layers, we use the 2D DWT and 2D IDWT to replace the traditional downsampling and upsampling. In the process of downsampling, we use the DWT to decompose the image information into low-frequency and high-frequency components and use the DCT to enrich the low-frequency information of the network. Then, the low-frequency components enter the New Receptive Field Block for feature aggregation so that the high-frequency components can be transmitted to the upsampling layer. During upsampling, the IDWT uses high-frequency components and aggregated features to restore data details and reconstruct the salient objects of the image.

Figure 2 shows the architecture of the proposed FESNet. In the stage of coding, an RGB image, after convolved, will enter the network layer composed of the DWT and DCT, to obtain the low-frequency and high-frequency information of the image. Then, a New Receptive Field Block (NRFB) will aggregate the features of the input low-frequency information. In the stage of decoding, the aggregated features and the two groups of high-frequency components output from the encoder will be upsampled by the inverse discrete wavelet transform (IDWT). Finally, the decoder uses the convolutional block to refine the feature map and reconstructs the image through bilinear interpolation. In Figure 2, RGB and GT, respectively, refer to the original input image and the pixel-wise labeled ground truth image. CB is a common Convolution Block with a receptive field of size 3 × 3.

As shown in Figure 3, we introduce the DCT in the bottleneck of each layer. After the DCT of the input data, the information will be concentrated in the low-frequency part. Thus, the network can have its low-frequency components effectively enriched.

We use the New Receptive Field Block in Figure 4 to aggregate image features and enlarge the receptive field for the small objects. Different from the conventional RFB module, we cascade the three dilated convolution layers in series, then aggregate them with other branches; we finally add the normal convolutional branch of the input feature of this block, and obtain the image features as a whole through the Relu function.

## 4. Experiments and Results

### 4.1. Datasets

We trained our network on GrainPest and SOC datasets. The dataset GrainPest contains 500 pixel-level labeled grain pest images. The grain pest images are made in the background of wheat, corn, rice, and other grains. The grain pests include *Sitophilus zeamais (Sz)*, *Sitotroga cerealella (Sc)*, *Rhizopertha dominica (Rd)*, *Plodia interpunctella (Pi)*, and other species of grain pest. We gathered the images by taking pictures at granaries and obtaining images from professional entomology web sites, such as Insect Images (https://www.insectimages.org/, accessed on 6 November 2021) and iNaturalist (https://www.inaturalist.org/, accessed on 18 December 2021). In the actual detection, a variety of grains were uninfected with pests, so the dataset GrainPest contains 50 non-salient object images with pure grains as the background. The dataset SOC contains 6000 images, including salient and non-salient images of more than 80 categories of daily objects. The attributes of non-salient and small objects in the GrainPest and SOC datasets bring more challenges to salient detection models than common datasets.

### 4.2. Evaluation Metrics

We adopt four evaluation metrics, Mean Absolute Error (MAE), F-measure, S-measure, and E-measure, to quantitatively evaluate the model performance; the evaluation toolbox can be found at https://github.com/DengPingFan/Evaluation-on-salient-object-detection, accessed on 9 March 2022.

MAE is mainly used to evaluate the pixel-level error between the generated salient map M and the ground truth G:(14)$$MAE=\frac{1}{W\times H}\sum_{x=1}^{W}\sum_{y=1}^{H}\left\|M(x,y)-G(x,y)\right\|$$
where 𝑊, and 𝐻 are the width and height of the image. A smaller MAE indicates a better performance. The F-measure, as a weighted summed average of precision and recall, has non-negative weights and the results are more reliable:(15)Fm=(1+β2)⋅Precision⋅Recallβ2⋅Precision+Recall
where $\beta^{2}$ is set to the threshold of 0.3 by previous experimental work, in order to tradeoff between precision and recall. A larger $F_{m}$ indicates a better performance. We choose the max F-measure in our paper. S-measure takes into account both object-aware (So) and region-aware (Sr) structural correlations:(16)$$S_{m}=\alpha \times S_{o}+(1-\alpha )\times S_{r}$$ where α is the proportion of the foreground area in the image, which is empirically set to 0.5%. A larger $S_{m}$ indicates a better performance. E-measure ($E_{m}$) measures the structural similarity between the prediction mask and the ground truth. We choose the max E-measure in our paper.
(17)$$E_{m}=\frac{1}{W\times Q}\sum_{i=1}^{W}\sum_{j=1}^{H}\phi s(i,j)$$
where ϕs is the enhanced alignment matrix, and H and W are the height and the width of the map.

### 4.3. Qualitative and Quantitative Results

The experiment was carried out on the Ubuntu 20.04 operating system, configured with an Nvidia GTX 1080Ti, cuda11.6, and cuDNN8.4.0. Pytorch 1.11.0 was adopted as the deep learning framework. All experiments followed the same settings.

As the visualization results show in Figure 5, four SOTA saliency detection models, a Deep Hierarchical Saliency Network (DHSNet) [36], a U^2^-Net [14], a Multi-Scale Difference of Gaussians Fusion in Frequency (MDF) [37], and Wavelet Integrated Deep Networks for Image Segmentation (WaveSNet) [38] were used in the experiment to compare with our FESNet. We completed a verification of the datasets (GrainPest and SOC). Table 1 shows the qualitative evaluation results of the different saliency detection models, and the proposed FESNet outperformed all the comparative models. On the dataset GrainPest, E-measure and F-measure of FESNet are 1.7% and 1.6%, respectively, higher than U^2^-Net. Especially on the dataset SOC, we achieved a 4% and 5.9% gain in E-measure and F-measure against the second best model, U^2^-Net. This demonstrates that our model can perform better in complex scenes. F-measure is the weighted harmonic mean of the recall rate R and precision P, and E-measure is the global average of the image and local pixel matching. The improvement of the two values intuitively shows the improvement in the accuracy of the model in detecting stored-grain pests. Among them, “DHSNet, U^2^-Net, MDF, and WaveSNet” are the abbreviations of the models’ names.

### 4.4. Ablation Study

To verify the effectiveness of our FESNet network, we carried out ablation research on the following three aspects: Firstly, to verify that the DCT module we proposed can effectively enhance low-frequency information in the frequency domain, we conducted ablation research by comparing the model data before and after the DCT was added. Table 2 shows the quantitative results from the ablation research. It can be seen that, after using the DCT module, our model achieves the best results.

Then, to compare the advantages of ASPP and the New Receptive Field Block in converging frequency-domain information, we carried out ablation research by setting a contrast experiment. Table 3 shows the comparative data obtained from the experiment. As we have seen, our NRFB can better enrich the detection accuracy of grain pests.

Finally, we used the New Receptive Field Block of different layers to verify its effectiveness and build a more expressive model by designing a contrast experiment. As the data shows in Table 4, we used the three-layer New Receptive Field Block to achieve better performance. All of our ablation research follows the same settings for implementation.

## 5. Conclusions

In this paper, we propose a frequency-enhanced saliency (FESNet) detection model resulting in the pixelwise segmentation of grain pests. The frequency clues, as well as the spatial information, are leveraged to enhance the detection performance of small-grain insects from the cluttered background. Firstly, we design the discrete wavelet transform (DWT) and the inverse discrete wavelet transform (IDWT) in the frequency domain as the general layer of the network, in order to acquire the high-frequency information of images. Secondly, we introduce a discrete cosine transform (DCT) into each layer of the network to enhance its low-frequency information by enhancing channel attention. Then, we propose a new receptive field block (NRFB) to enlarge the receptive fields by aggregating three atrous convolution features. Finally, in the phase of decoding, we use the high-frequency information and aggregated features together to restore the saliency map. Simultaneously, we construct a saliency detection dataset, GrainPest, especial for stored-grain pests. This dataset comes from a real storage scene with a cluttered environment, and the images are composed of grains and pests. Considering that plenty of grains are uninfected with pests in the actual detection, we added pure grain images with non-salient objects to enhance the generalization and robustness of the proposed model. Our solution incorporated new techniques of frequency convolution to explore the tasks of small insects and non-salient object segmentation. The experimental results show that the proposed FESNet achieved a more favorable performance in all metrics than the SOTA models. We will further promote our research on the segmentation of insects and simultaneously offer species identification and semantic segmentation for adoption of different approaches in order to control different grain insects. The benchmark dataset for the saliency detection of grain pests, proposed in this paper, will be published online for reference and comparison by researchers in this field.

## Figures and Tables

**Figure 1 insects-14-00099-f001:**
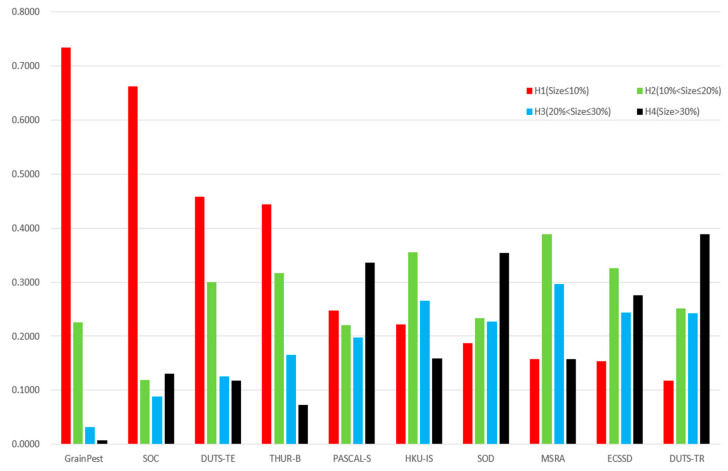
Analysis of the proportion of salient objects in the popular datasets. The red bars represent the proportion of small targets in the dataset. It can be seen that the small targets in the GrainPest and SOC datasets account for more than half, and the ECCSD and DUTS-TR data sets are mostly medium or large targets.

**Figure 2 insects-14-00099-f002:**
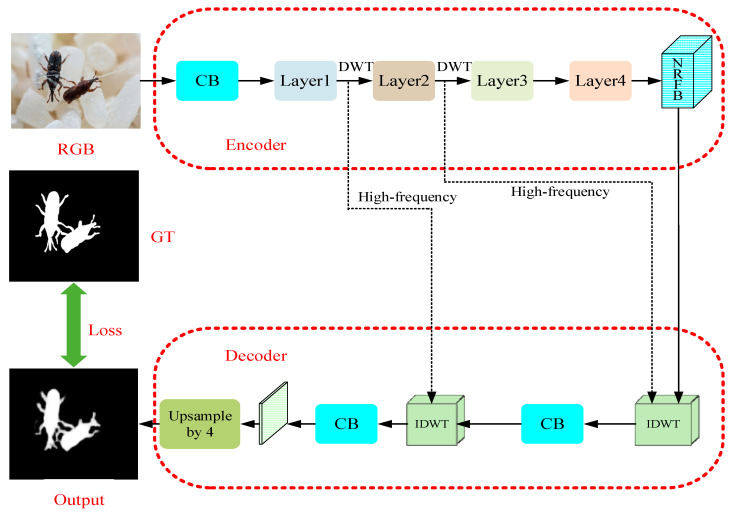
The architecture of the proposed FESNet. The main architecture is a DeeplabV3+-like Encoder–Decoder, where each stage consists of a DWT/IDWT layer and a DCT channel attention module. The detailed configuration of each stage is illustrated in Figure 3 and Figure 4.

**Figure 3 insects-14-00099-f003:**
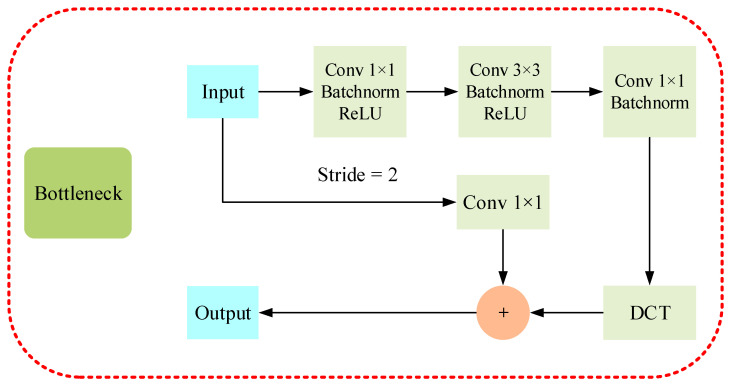
Bottleneck with discrete cosine transforms added. After the data is subjected to three Conv Blocks, the low-frequency information is enriched by the DCT. Inclusively, when Strip is 2, it will perform the Conv operation.

**Figure 4 insects-14-00099-f004:**
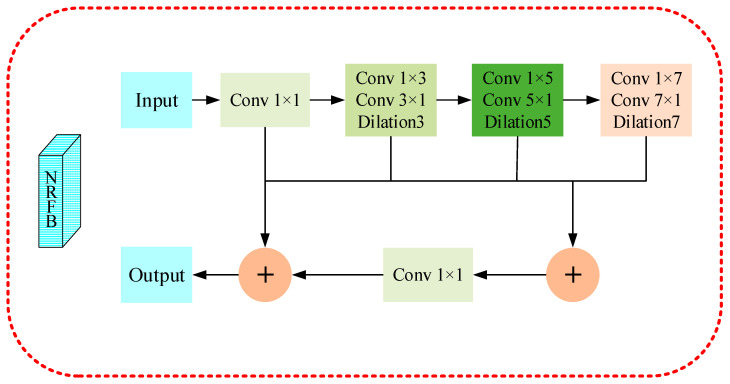
Illustration of our proposed New Receptive Field Block (NRFB). This module first reduces the channel dimension of the high-level output features of the backbone network through 1×1 Conv and then uses three groups of dilation convolution layers to enlarge the receptive field.

**Figure 5 insects-14-00099-f005:**
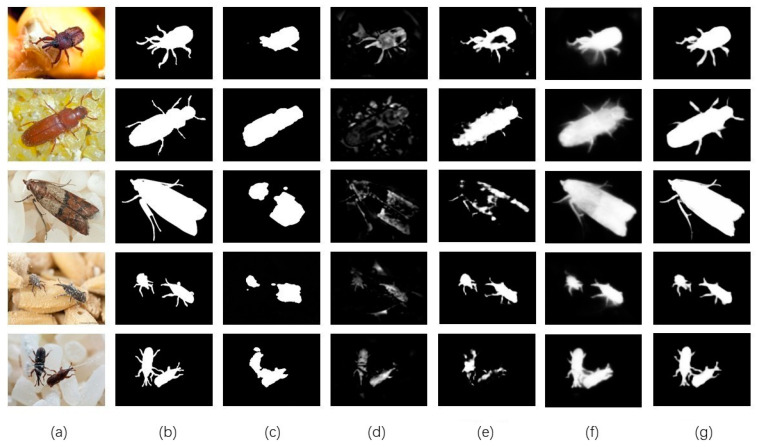
Qualitative comparison of the proposed method with four SOTA saliency detection methods: (**a**) image, (**b**) GT, (**c**) DHSNet, (**d**) MDF, (**e**) WaveSNet, (**f**) U^2^-Net, (**g**) Ours.

**Table 1 insects-14-00099-t001:** Evaluation of state-of-the-art models on GrainPest and SOC. The best and second-best performances are highlighted in **red** and green.

Method	GrainPest				SOC			
	$S_{m}↑$	MAE↓	$E_{m}↑$	$F_{m}↑$	$S_{m}↑$	MAE↓	$E_{m}↑$	$F_{m}↑$
DHSNet	0.819	0.031	0.889	0.671	0.800	0.122	0.848	0.289
MDF	0.659	0.068	0.870	0.512	0.699	0.130	0.746	0.198
WaveSNet	0.722	0.048	0.863	0.543	0.716	0.137	0.727	0.211
U^2^-Net	0.899	** 0.024 **	0.953	0.742	0.828	0.095	0.871	0.309
FESNet	** 0.903 **	** 0.024 **	** 0.970 **	** 0.758 **	** 0.872 **	** 0.064 **	** 0.911 **	** 0.368 **

**Table 2 insects-14-00099-t002:** Ablation studies of the DCT module. Red indicates the best performance.

Contrast Module	GrainPest			
	$S_{m}↑$	MAE↓	$E_{m}↑$	$F_{m}↑$
noDCT	0.889	0.029	0.954	0.725
DCT	0.903	0.024	0.970	0.758

**Table 3 insects-14-00099-t003:** Ablation studies of ASPP and NRFB modules. Red indicates the best performance.

Contrast Module	GrainPest			
	$S_{m}↑$	MAE↓	$E_{m}↑$	$F_{m}↑$
ASPP	0.888	0.031	0.959	0.739
NRFB	0.903	0.024	0.970	0.758

**Table 4 insects-14-00099-t004:** Ablation studies of NRFB modules with different cascades. Red indicates the best performance.

Contrast Module	Channels	GrainPest			
		$S_{m}↑$	MAE↓	$E_{m}↑$	$F_{m}↑$
First-level NRFB	2048→256	0.897	0.027	0.968	0.746
Two-level NRFB(a)	2048→1024→256	0.903	0.024	0.967	0.753
Two-level NRFB(b)	2048→512→256	0.893	0.030	0.958	0.744
Three-level NRFB	2048→1024→512→256	0.903	0.024	0.970	0.758

## Data Availability

The data that support the findings of this study are available at https://pan.baidu.com/s/1aNuHF8YJ5BjmvKhXL7rUsA, accessed on 20 November 2022. The extraction code are available on request from the corresponding author by yujunwei@126.com.

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
