# Peer review of "FESNet: Frequency-Enhanced Saliency Detection Network for Grain Pest Segmentation"

_insects, 2023, doi:10.3390/insects14020099_

Round 1

Reviewer 1 Report

In this manuscript, an enhanced frequency detection model (FESNet) is proposed, which results in the segmentation of grain pests by pixels, which could significantly contribute to the reduction of losses in stored grain from storage pests. Therefore, I suggest publishing this manuscript in a special issue of this journal.

Author Response

Thanks for the reviewer's comments. We have studied all the comments carefully and improved our paper with your helpful suggestions.

Reviewer 2 Report

The author introduced FESNet for grain pests segmentation and with the proposed dataset they gave a significant contribution in grain pest detection and monitoring. The research is well designed, and the each part of the manuscript is supported by recent relevant references. Still, some minor issue should be improved; 

Line 139 Does the reference Qin et al. (15) refer to the previous statement? It is not clear form the way it is stated.

Line 286, 287 Use Italic font for all the names of insect species that is mentioned. Change Rhizopertha Dominica with Rhyzopertha dominica

Can the proposed dataset Grain Pest determine the pest to the insect species? It is relevant, because the identification of species that belong to primary insect group (both Coleoptera and Lepidoptera species) requires different approach in their control compared to secondary group of pests. Author should appoint that within the conclusion or where it is appropriate.

Reviewer 3 Report

Comments

on the article  "FESNet: Frequency-Enhanced Saliency Detection Network for Grain Pests Segmentation " by Junwei Yu, Fupin Zhai, Nan Liu, Yi Shen and Quan Pan, submitted to the "Insects".

The subject of the reviewed paper is significant. Today it is obvious that solving the global food crisis and improving food security depend not only on increasing staple food production but also on the prevention/reduction of post-harvest losses, which are mainly caused by insect pests. Systematic monitoring for stored product insect pests, one of the significant biotic risk factors in grain storage, is a critical component in the integrated pest management of stored grain. Therefore, detecting insect pests and estimating their population density becomes necessary and even crucially impotent for making proper decisions for insect infestation control. Today, the primary method for grain insect pest monitoring is probe sampling, which requires a lot of manual labor and time, and does not satisfy modern grain storage systems. In the last few years, some publications on developing the image recognition method for the stored product insect pest were published. The advantages of this method include rapid and accurate grain insect pest detection. The reviewed article is devoted to developing an improved model for grain insect detection and is worth publication.

However, there are some comments:

1. Line 68-69: "A lot of image-based methods are proposed for the detection ... of grain pests." Only two examples are cited. 

2. Line 163-165: "... most object detection models ..., such as SOC... contain only one or two large salient targets ... ". The figure below shows that at least 65% of SOC images contain small objects.  

3. Figure 1: The legend is unclear. Only "red bars" are described in the figure caption. The reader has to guess that other colors represent larger objects. The following text in lines 180-183 defines the H1-H4 metric, so it should come before the figure.

4. General to subsection 2.3: The section is supposed to describe the "related work" (as the whole section 2 describes) of datasets with salient objects. However, the central part of the section focuses on the new GrainPest dataset and its advantages over existing datasets in terms of the proportion of images with small objects. Moreover, the cited datasets (like SOC, DUTS-TR, etc.) are general image datasets, not specific to pests. It is not surprising that these datasets are not focusing on small objects. Has any comparison to pests-images datasets been made? 

5. Figure 2: The notations CB and GT on the diagram are unclear. It may be obvious to an image-processing expert but not to a general reader of Insects

6. Line 292-293: "The dataset is closer to the real world scene and more challenging for salient object detection ...".   More challenging than what dataset? 

7. Subsection 4.2: The rationale for choosing the parameters' values like betta-squared and alpha is not described enough and remains unclear. No mathematical definition is presented for E-measure (in contrast to F and S measures).

8. The subsections "4.1. Datasets" and "4.2. Evaluation metrics" do not consistent with the title of the section "Experiments and Results" and, in my opinion, should be moved to the section "Material and Methods".

9. Subsection 4.3 is titled "Implementation details", but only the first paragraph describes the implementation. The following paragraph presents the results.

10. Line 318-319: "As can be seen ... 1.7% and 1.6% respectively higher ... ". It is not clear how important such a relatively small increase in E and F-measures is. It may be significant, but for a general reader, it is not clear.

11. The same comment for the Ablation study (subsection 4.4). In this subsection, some slight performance improvement is presented, and it is unclear if the effort is worth it. 

12. Table 1. "Red and green indicate the best and the second best performance".  However, there are no red and green marks on the table.

13. General comment to section 4. There are no computational performance measures, like total run-time for output. Is it not necessary in practical applications? For the specific HW and SW setting mentioned in section 4.3, it may be interesting to compare DHSNet run-time performance to other methods.

14. Conclusions. The section lacks conclusions about the proposed method's performance compared to other methods. How good is it? What benefit may it bring to practical applications?  

15. As the article is submitted to the Insects Journal, some specific physics and mathematics terms and abbreviations, such as "GT, CB, RSU, DHSNet, U2-Net, MDF, and WaveSNet", should be explained, not only by references.

16. The online publication of the dataset for the saliency detection of grain pests before or in parallel with the paper publishing seems to be more effective for reference and comparison by researchers in this field.

17. Some references (7-11, 14, 15, 22, 26, 28, 32, 34) as well as grammar throughout the whole text, should be corrected.

After accepting these comments or after the motivated author's answer, the article in the new version may be published.
